# Disparities in telemedicine utilization among surgical patients during COVID-19

Courtney M. Lattimore[1], William J. Kane[1], Mark A. Fleming, II[1], Allison N. Martin[2], J. Hunter Mehaffey[1], Mark E. Smolkin[3], Sarah J. Ratcliffe[3], Victor M. Zaydfudim[1], Shayna L. Showalter[1], Traci L. Hedrick[1]*

1 Department of Surgery, University of Virginia Health System, Charlottesville, Virginia, United States of America, 2 Department of Surgical Oncology, The University of Texas MD Anderson Cancer Center, Houston, Texas, United States of America, 3 Division of Biostatistics, Department of Public Health Sciences, School of Medicine, University of Virginia, Charlottesville, Virginia, United States of America

* th8q@hscmail.mcc.virginia.edu

## Abstract

### Background

Telemedicine has been rapidly adopted in the wake of the COVID-19 pandemic. There is limited work surrounding demographic and socioeconomic disparities that may exist in telemedicine utilization. This study aimed to examine demographic and socioeconomic differences in surgical patient telemedicine usage during the COVID-19 pandemic.

### Methods

Department of Surgery outpatients seen from July 1, 2019 to May 31, 2020 were stratified into three visit groups: pre-COVID-19 in-person, COVID-19 in-person, or COVID-19 telemedicine. Generalized linear models were used to examine associations of sex, race/ethnicity, Distressed Communities Index (DCI) scores, MyChart activation, and insurance status with telemedicine usage during the COVID-19 pandemic.

### Results

14,792 patients (median age 60, female [57.0%], non-Hispanic White [76.4%]) contributed to 21,980 visits. Compared to visits before the pandemic, telemedicine visits during COVID-19 were more likely to be with patients from the least socioeconomically distressed communities (OR, 1.31; 95% CI, 1.08,1.58; $P$ = 0.005), with an activated MyChart (OR, 1.38; 95% CI, 1.17–1.64; $P$ < .001), and with non-government or commercial insurance (OR, 2.33; 95% CI, 1.84–2.94; $P$ < .001). Adjusted comparison of telemedicine visits to in person visits during COVID-19 revealed telemedicine users were more likely to be female (OR, 1.38, 95% CI, 1.10–1.73; $P$ = 0.005) and pay with non-government or commercial insurance (OR, 2.77; 95% CI, 1.85–4.16; $P$ < .001).

### Conclusions

During the first three months of the COVID-19 pandemic, telemedicine was more likely utilized by female patients and those without government or commercial insurance compared

**Data Availability Statement:** All relevant data are within the manuscript and its Supporting information files.

**Funding:** Research reported in this article was supported by award T32CA163177 (CML and WJK) from the National Institutes of Health, the University of Virginia Department of Surgery. The funders had no role in study design, data collection and analysis, decision to publish, or preparation of the manuscript.

**Competing interests:** The authors have declared that no competing interests exist.

to patients who used in-person visits. Interventions using telemedicine to improve health care access might consider such differences in utilization.

## Introduction

The COVID-19 pandemic swiftly upended life as we know it with the decline in social, economic, mental, and physical wellbeing which was felt around the world. As hospitals braced for overwhelming COVID-19 patient admissions while ensuring an adequate supply of personal protective equipment (PPE), health systems were tasked with finding a way to safely triage, evaluate, and treat patients in accordance with national COVID-19 safety guidelines. Telemedicine, a previously underutilized virtual platform in which patients and providers are able to communicate via telephone or videoconference, became essential; offering a safe and convenient alternative care delivery platform amidst the pandemic [1, 2]. As much as a 50–300 fold increase in the number of patients evaluated by telemedicine during the pandemic has been observed [3, 4]. At least 50% of physicians have transitioned to incorporating telemedicine into their practice, most of which have never used this platform previously [1, 2, 4].

This rapid change in healthcare delivery has raised concerns regarding which patients can access healthcare under conditions of a major public health crisis. Unsurprisingly, the COVID-19 pandemic has further exposed healthcare disparities that exist in the United States [5–7]. Studies of COVID-19 have shown that Black and Hispanic communities are disproportionately impacted by the virus [5, 6, 8, 9], and are more likely to contract COVID-19, become hospitalized, and die [8, 9]. Patients who are elderly, of low socioeconomic status, and suffer from pre-existing health conditions are also at particularly increased risk [5, 7, 8]. These vulnerable populations could greatly benefit from access to quality care in the form of telemedicine.

Telemedicine is a convenient form of communication and improves access to care for those with the required technology, but can exacerbate disparities in those who do not. Video visits are the exclusive form of telecommunication in some practices because they not only offer a visual evaluation of the patient that is otherwise lost over the phone, but also reimburse at a higher rate [10]. However, video platforms require a stable internet connection, updated electronics with video capabilities, and proficient digital literacy [4, 10]—resources that may be inadequate for rural, minority, elderly, non-English-speaking, and low socioeconomic patients [7, 10]. Clinician factors such as poor cultural competency and discomfort with technology can also make telemedicine particularly difficult for these populations [4, 7, 11]. Moreover, minority and low socioeconomic status populations are historically more distrustful of the healthcare system due to under investigated mechanisms of systemic racism, which can be exacerbated in a major public health crisis [12, 13]. These populations, which are already less likely to seek medical care in traditional ways, may be even less likely to choose and trust telemedicine, a more novel avenue of receiving care [14].

Investigation and understanding of access, barriers, and disparities in telemedicine utilization during the COVID-19 pandemic is essential to providing equitable heath care and the future implementation of telemedicine into medical practice. We sought to answer whether the rapid adoption of telemedicine has created additional disparities in access to health care for surgical patients in vulnerable, at-risk populations. We hypothesized that during the COVID-19 pandemic, telemedicine in a surgical population was disproportionately used based on race, sex, and socioeconomic status.

## Methods

### Study population

All billable Department of Surgery outpatient office and telemedicine encounters at a single academic tertiary care center from July 1, 2019 to May 31, 2020 were included in the study, which included new patient visits and follow up visits. Non-billable postoperative visits during the global period were not included within this particular database. On March 17, 2020, institutional leadership mandated rescheduling of all non-urgent outpatient visits, signaling the beginning of the COVID-19 pandemic in our community. All encounters in this study were divided into one of three groups: visits from July 1, 2019 to March 17, 2020 (Pre-COVID-19), in-person visits on or after March 17, 2020 (COVID-19 in-person), or telemedicine visits on or after March 17, 2020 (COVID-19 telemedicine).

### Study variables

Encounter data were extracted from prospectively maintained institutional databases. Specific variables captured for each visit included patient age, sex, race/ethnicity as defined by the National Institute of Health (non-Hispanic White, non-Hispanic Black, Hispanic, and Other/ Unknown) [15], language preference, visit payer (e.g., Medicare, Medicaid, private insurer, and Other: self-pay, financial assistance, other means). MyChart activation status, home ZIP code, and subspecialty, which was based on the clinician/clinic the patient was evaluated. MyChart is an internet or smartphone application-based portal for patients to access their medical record and communicate with clinicians. MyChart was the method of choice for contacting patients regarding their telemedicine visit and granting access to a third-party video platform. For telemedicine visits, the mode of communication—i.e., video or telephone—was also extracted from an institutional database or, when unknown, obtained from review of the electronic medical record.

Patient home ZIP code was used to calculate two additional variables: driving distance to the main hospital clinic and the patient's Distressed Communities Index (DCI) score. DCI is a composite measure of community-level socioeconomic distress ranging from 0 (least distress) to 100 (most distress) and has been well-documented in the surgical literature [9, 16]. Patient DCI scores were grouped based on nationally determined quintiles (i.e. distressed, at risk, mid-tier, comfortable, or prosperous) consistent with prior literature [9, 16] and then further categorized as lower-tier (distressed and at risk), mid-tier (mid-tier), and top-tier (comfortable and prosperous).

Time period of patient visits were divided into five groups based on two-week intervals: March 17th-31st, April 1st-15th, April 16th-30th, May 1st-15th, and May 16th-31st.

### Statistical analysis

Summary statistics for the entire patient cohort as well as for each encounter group (i.e., Pre-COVID-19, COVID-19 in-person, or COVID-19 telemedicine) were calculated and are reported as frequencies with percentages or medians with interquartile ranges as appropriate. Demographics at the patient level are reported to highlight static characteristics (e.g. age, sex, race/ethnicity, driving distance, DCI score, language preference), while other characteristics that are subject to change are provided at the visit-level (e.g. provider subspecialty and MyChart activation status). Missing data was present for DCI score/group and driving distance, however represented < 5% of all patient-visits. Thus, a complete-case analysis was conducted.

The primary objective of this study was to identify patient and surgery clinic characteristics associated with completion of a telemedicine visit during COVID-19 compared to visits before COVID-19. Generalized estimating equations (GEEs) with a logit link and a robust, unstructured correlation structure were used to test for differences between groups. GEEs were chosen to produce population-level estimates, allowing for marginal odds ratios and predicted probabilities to be calculated, while adjusting for patients with more than one visit. Additionally, secondary objectives used similar methods to identify factors associated with completion of an in-person visit during COVID-19 compared to visits before COVID-19, as well as factors associated with completion of a telemedicine visit during COVID-19 compared to in-person visits during COVID-19. This latter comparison also adjusted for calendar time.

Due to concerns about modeling the age variable and the pediatric surgery category together, a sensitivity analysis that excluded pediatric patients was performed, yielding no appreciable differences in the model estimates.

A two-sided $P < 0.05$ was considered significant, with the results of secondary analyses considered exploratory. SAS version 9.4 (SAS Institute Inc.) was used to conduct all analyses. The University of Virginia Institutional Review Board (IRB) for Health Sciences Research reviewed and approved this study (IRB# 22462) and deemed it as meeting exempt determination. The IRB determined this to be secondary research in which informed consent is not required. Identifiable private information was collected, secured on an approved server, and deidentified for study analysis.

## Results

The study cohort included 14,792 patients contributing to 21,980 visits. A majority of the patients were female (57.0%) and non-Hispanic White (76.4%) with a median age of 60 years. The other race/ethnicities represented were non-Hispanic Black (14.6%), Hispanic (3.7%), and Other/Unknown (5.3%). Seventy-two percent of patients had just one visit during the study period while 18% had two visits and 10% had three or more visits.

Of the 21,980 visits, 19,407 (88.3%) were in-person and took place before COVID-19, 1,809 (8.2%) were in-person during COVID-19, and 764 (3.5%) were conducted via telemedicine during COVID-19 (Table 1). Telemedicine visits were primarily conducted via videoconference (67.5%). Substantial changes in the proportions of visits provided by each subspecialty were seen between visits before COVID, in-person visits during COVID, and telemedicine visits during COVID.

Adjusted analysis, accounting for all of the variables in the model, compared telemedicine visits during COVID-19 to visits before COVID-19 and found that telemedicine visits were more likely used by patients with an activated MyChart status (OR, 1.38; 95% CI, 1.17–1.64; $P$ < .001) and who were expected to self-pay, utilize financial assistance, or pay with other means for their visit, compared to those covered by Medicare (OR, 2.33; 95% CI, 1.84–2.94; $P$ < .001) (Table 2). Further, patients residing in a top-tier community, as compared to a low-tier community, were more likely to utilize a telemedicine visit (OR, 1.31; 95% CI, 1.08,1.58; $P$ = 0.005). Finally, surgical subspecialty was strongly associated with use of telemedicine during COVID, with significant variability across subspecialties (Fig 1). For example, patients evaluated in the Transplant Surgery clinic had a 6.9% chance of undergoing a telemedicine visit in the COVID period, while patients seen in Pediatric Surgery had only a 1.4% chance of having a telemedicine visit. A sensitivity analysis excluding pediatric patients showed no appreciable differences in the estimates obtained for this primary comparison (S1–S3 Tables).

Exploratory analysis of in-person visits during COVID-19, compared to in-person visits before COVID-19, showed that patients who underwent a visit during COVID-19 were more

**Table 1. Patient characteristics by visit type[a].**

| Characteristic | All Visits N = 21,980 | Pre-COVID-19 In-Person N = 19,407 | COVID-19 In-Person N = 1,809 | COVID-19 Telemedicine N = 764 |
|---|---|---|---|---|
| Age, y, median (IQR) | 60.0 (46.0–70.0) | 60.0 (46.0–70.0) | 59.0 (45.0–68.0) | 60.0 (48.0–69.0) |
| Sex | | | | |
| Female | 12,644 (57.5%) | 11,202 (57.7%) | 980 (54.2%) | 462 (60.5%) |
| Male | 9,336 (42.5%) | 8,205 (42.3%) | 829 (45.8%) | 302 (39.5%) |
| Race/Ethnicity | | | | |
| Non-Hispanic White | 16,557 (75.3%) | 14,663 (75.6%) | 1,331 (73.6%) | 563 (73.7%) |
| Non-Hispanic Black | 3,520 (16.0%) | 3,091 (15.9%) | 291 (16.1%) | 138 (18.1%) |
| Hispanic | 823 (3.7%) | 731 (3.8%) | 75 (4.1%) | 17 (2.2%) |
| Other/Unknown | 1,080 (4.9%) | 922 (4.8%) | 112 (6.2%) | 46 (6.0%) |
| English language preferred | 21,268 (96.8%) | 18,774 (96.7%) | 1,747 (96.6%) | 747 (97.8%) |
| Payer | | | | |
| Medicaid | 29,49 (13.4%) | 2,615 (13.5%) | 254 (14.0%) | 80 (10.5%) |
| Medicare | 9,395 (42.7%) | 8,359 (43.1%) | 745 (41.2%) | 291 (38.1%) |
| Private | 7,827 (35.6%) | 6,911 (35.6%) | 664 (36.7%) | 252 (33.0%) |
| Other | 1,809 (8.2%) | 1,522 (7.8%) | 146 (8.1%) | 141 (18.5%) |
| DCI Score, median (IQR)[b] | 52.2 (29.5–69.9) | 52.2 (29.6–69.9) | 52.2 (27.2–69.9) | 48.7 (27.2–69.0) |
| DCI Group[b] | | | | |
| Top-tier | 8,123 (37.8%) | 7,134 (37.6%) | 673 (38.1%) | 316 (42.2%) |
| Mid-tier | 5,449 (25.3%) | 4,813 (25.3%) | 445 (25.2%) | 191 (25.5%) |
| Lower-tier | 79,31 (36.9%) | 7,041 (37.1%) | 648 (36.7%) | 242 (32.3%) |
| Distance to clinic, mi, median (IQR)[c] | 52.3 (28.6–87.4) | 52.3 (28.5–87.1) | 52.3 (29.0–88.1) | 56.3 (31.5–103.5) |
| MyChart Activated | 12,093 (55.0%) | 10,551 (54.4%) | 1,073 (59.3%) | 469 (61.4%) |
| Communication Type | | | | |
| Video | | | | 516 (67.5%) |
| Phone | | | | 245 (32.1%) |
| Unknown | | | | 3 (0.4%) |
| Specialty | | | | |
| Breast | 3562 (16.2%) | 3163 (16.3%) | 251 (13.9%) | 148 (19.4%) |
| Colorectal | 1930 (8.8%) | 1757 (9.1%) | 129 (7.1%) | 44 (5.8%) |
| Cardiothoracic | 2815 (12.8%) | 2493 (12.8%) | 206 (11.4%) | 116 (15.2%) |
| General | 1220 (5.6%) | 1086 (5.6%) | 90 (5.0%) | 44 (5.8%) |
| MIS/Bariatric | 2936 (13.4%) | 2646 (13.6%) | 232 (12.8%) | 58 (7.6%) |
| Oncology | 2289 (10.4%) | 1961 (10.1%) | 229 (12.7%) | 99 (13.0%) |
| Pediatric | 1162 (5.3%) | 1048 (5.4%) | 100 (5.5%) | 14 (1.8%) |
| Transplant | 2830 (12.9%) | 2321 (12.0%) | 313 (17.3%) | 196 (25.7%) |
| Vascular | 3236 (14.7%) | 2932 (15.1%) | 259 (14.3%) | 45 (5.9%) |

[a] Patients with multiple visits will appear more than once in this table, thus inflating their characteristics.

[b] DCI score/group only available on n = 21,503 visits.

[c] Distance only available on n = 21,062 visits.

DCI = distressed communities index, MIS = minimally invasive surgery.

likely to be of younger age (OR, 1.00; 95% CI, 0.99–1.00; $P = 0.043$), have activated MyChart (OR, 1.29; 95% CI, 1.14–1.45; $P < .001$), and less likely to self-pay, utilize financial assistance, or pay with other means for their visit, compared to those covered by Medicare (OR, 0.75; 95% CI, 0.57–0.98; $P = 0.038$) (Table 3). Similarly, exploratory analysis of only visits during

**Table 2. Patient characteristics associated with a telemedicine visit during COVID-19, compared to pre-COVID in-person visits.**

| Variable | Odds Ratio | 95% Confidence Limits | |
|---|---|---|---|
| Age, y | 1.00 | 0.99 | 1.01 |
| Female | 1.17 | 0.99 | 1.40 |
| Race/Ethnicity | | | |
| Non-hispanic White | 1 [Ref] | | |
| Non-hispanic Black | 1.11 | 0.89 | 1.38 |
| Hispanic | 0.65 | 0.36 | 1.18 |
| Other/Unknown | 1.37 | 0.97 | 1.94 |
| English language preferred | 1.76 | 0.96 | 3.23 |
| Payer | | | |
| Medicare | 1 [Ref] | | |
| Medicaid | 1.16 | 0.84 | 1.60 |
| Private | 1.01 | 0.81 | 1.25 |
| Other | 2.33 | 1.84 | 2.94 |
| DCI Group | | | |
| Top-tier | 1 [Ref] | | |
| Mid-tier | 0.95 | 0.77 | 1.16 |
| Lower-tier | 0.77 | 0.63 | 0.92 |
| log(Distance, mi) | 1.01 | 0.92 | 1.12 |
| MyChart Activated | 1.38 | 1.17 | 1.64 |
| Specialty | | | |
| Breast | 1.10 | 0.75 | 1.60 |
| Colorectal | 0.66 | 0.42 | 1.02 |
| Cardiothoracic | 1.23 | 0.85 | 1.80 |
| General | 1 [Ref] | | |
| MIS/Bariatric | 0.56 | 0.37 | 0.86 |
| Oncology | 1.38 | 0.93 | 2.04 |
| Pediatric | 0.36 | 0.18 | 0.72 |
| Transplant | 1.85 | 1.27 | 2.68 |
| Vascular | 0.39 | 0.24 | 0.61 |

DCI = distressed communities index, MIS = minimally invasive surgery

COVID-19 showed that patients who underwent a telemedicine visit, compared to those who underwent an in-person visit, were more likely to be female (OR, 1.38, 95% CI, 1.10–1.73; $P = 0.005$) and pay for their visit with other, non-government or commercial insurance means (OR, 2.77; 95% CI, 1.85–4.16; $P < .001$) (Table 4).

## Discussion

The COVID-19 pandemic has highlighted the stark disparities that exist in marginalized communities [5, 7, 17]. There are numerous studies examining these inequities and some more recent studies investigating these in the context of telemedicine amidst the pandemic [18, 19]. However, the present study examined the adoption of telemedicine across a large general surgery population at a tertiary center during the COVID-19 pandemic, and more importantly, the characteristics of the patient population most likely to use telemedicine. Overall, telemedicine was more likely utilized by those without government or commercial insurance and women.

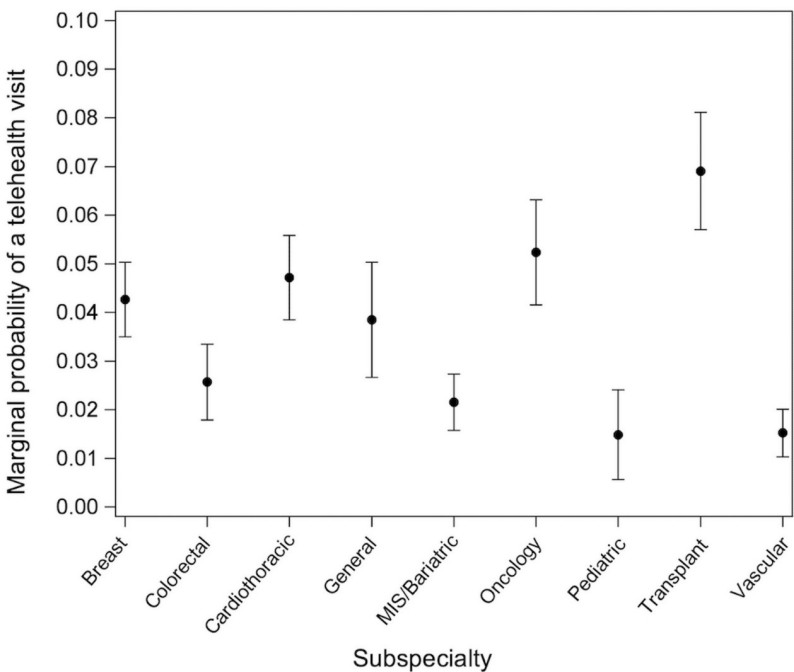

**Fig 1. Probability of telemedicine visit during COVID-19 by surgical subspecialty.** Marginal probability of a telemedicine visit during COVID-19, compared to visits before COVID-19, by surgical subspecialty.

The finding that female patients were more likely to use telemedicine versus in-person visits is consistent with other published literature on telemedicine use both prior to and during the pandemic [2, 20–22]. A recent cross-sectional study of 1.1 million patients found that men were significantly less likely to choose both a video and phone visit [22]. Similarly, a survey study determining preference and perceptions of telemedicine in an outpatient clinic setting also found predictors of liking telemedicine to be skewed towards female patients [20]. The pandemic has exacerbated responsibilities for women tasked with balancing career, childcare, and other caregiving/household family needs [20, 23], which could also be driving the higher rates of telemedicine use among women during this time.

In this study, there was a significant disparity in telemedicine use between patients who reside in communities of low compared to high socioeconomic distress. Those from the most distressed communities were significantly less likely to use telemedicine visits during COVID-19 compared to pre-COVID in-person visits. Several studies have identified factors that support this finding. In a recent survey study, patients with a household income of ≥ $50,000 were 34% more likely to choose a telemedicine visit than those making < $50,000 [24]. Additionally, although mobile devices are used by patients of all socioeconomic statuses, devices without video capability and lack of access to a stable internet connection can be a major barrier in low-income populations and may explain the preference for in-person and telephone visits over video [7, 22]. Patients of low socioeconomic backgrounds are less likely to have general digital literacy, access to broadband internet, and own a smartphone [4]. Furthermore, patients who live in areas with higher residential internet access have been found to be more likely to choose telemedicine [22]. The present study further emphasizes the importance of community environment and socioeconomic status in access to telemedicine.

Recognizing the disparities that exist across the country in broadband internet access, the Federal Communications Commission (FCC) has made a push over the past few years to

**Table 3. Patient characteristics associated with an in-person visit during COVID-19, compared to in-person visits before COVID-19.**

| Variable | Odds Ratio | 95% Confidence Limits | |
|---|---|---|---|
| Age, y | 1.00 | 0.99 | 1.00 |
| Female | 0.92 | 0.81 | 1.04 |
| Race/Ethnicity | | | |
| Non-hispanic White | 1 [Ref] | | |
| Non-hispanic Black | 0.99 | 0.84 | 1.17 |
| Hispanic | 1.27 | 0.88 | 1.84 |
| Other/Unknown | 1.41 | 1.11 | 1.80 |
| English language preferred | 0.96 | 0.65 | 1.42 |
| Payer | | | |
| Medicare | 1 [Ref] | | |
| Medicaid | 0.98 | 0.80 | 1.22 |
| Private | 0.92 | 0.79 | 1.08 |
| Other | 0.75 | 0.57 | 0.98 |
| DCI Group | | | |
| Top-tier | 1 [Ref] | | |
| Mid-tier | 1.02 | 0.88 | 1.18 |
| Lower-tier | 1.01 | 0.88 | 1.16 |
| log(Distance, mi) | 0.97 | 0.90 | 1.04 |
| MyChart Activated | 1.29 | 1.14 | 1.45 |
| Specialty | | | |
| Breast | 0.93 | 0.70 | 1.23 |
| Colorectal | 0.80 | 0.59 | 1.09 |
| Cardiothoracic | 1.02 | 0.77 | 1.35 |
| General | 1 [Ref] | | |
| MIS/Bariatric | 1.14 | 0.87 | 1.50 |
| Oncology | 1.44 | 1.09 | 1.89 |
| Pediatric | 0.93 | 0.64 | 1.35 |
| Transplant | 1.72 | 1.28 | 2.32 |
| Vascular | 0.96 | 0.73 | 1.27 |

DCI = distressed communities index, MIS = minimally invasive surgery

invest in infrastructure to provide access to vulnerable communities [25]. Although access has improved, it remains inadequate and highly variable in quality depending on location [25]. In Virginia, highly dense areas such as Northern Virginia benefit from investment of private companies in broadband coverage while less dense, rural areas are left uncovered [26]. There are at least 660,000 households and businesses without any broadband access and even more with inadequate or inaccessible coverage [26]. During the pandemic, several states, including Virginia, used the Coronavirus Relief Fund (CRF), as part of the Coronavirus Aid, Relief, and Economic Security (CARES) Act, to emergently provide broadband access to their communities as distanced work, education, and healthcare became a necessity [27, 28]. Unfortunately, this is not a permanent fixture [27] and if we want to strive towards equitable telemedicine access in the future, advocating for policy change will be essential.

Access to the internet has also been shown to be associated with MyChart activation [29], corroborating the result of this study that MyChart activation was associated with a 40% greater likelihood to have a telemedicine visit during COVID-19 compared to an in-person

**Table 4. Patient characteristics associated with a telemedicine visit during COVID-19, compared to an in-person visit during COVID-19.**

| Variable | Odds Ratio | 95% Confidence Limits | |
|---|---|---|---|
| Age, y | 1.01 | 1.00 | 1.02 |
| Female | 1.38 | 1.10 | 1.73 |
| Race/Ethnicity | | | |
| Non-hispanic White | 1 [Ref] | | |
| Non-hispanic Black | 1.16 | 0.88 | 1.54 |
| Hispanic | 0.40 | 0.15 | 1.04 |
| Other/Unknown | 0.99 | 0.64 | 1.53 |
| English language preferred | 1.22 | 0.48 | 3.10 |
| Payer | | | |
| Medicare | 1 [Ref] | | |
| Medicaid | 1.07 | 0.73 | 1.56 |
| Private | 1.08 | 0.82 | 1.42 |
| Other | 2.77 | 1.85 | 4.16 |
| DCI Group | | | |
| Top-tier | 1 [Ref] | | |
| Mid-tier | 0.93 | 0.72 | 1.20 |
| Lower-tier | 0.80 | 0.63 | 1.02 |
| log(Distance, mi) | 1.06 | 0.93 | 1.21 |
| MyChart Activated | 1.15 | 0.93 | 1.42 |
| Specialty | | | |
| Breast | 1.26 | 0.79 | 2.01 |
| Colorectal | 0.90 | 0.52 | 1.55 |
| Cardiothoracic | 1.62 | 0.99 | 2.65 |
| General | 1 [Ref] | | |
| MIS/Bariatric | 0.64 | 0.39 | 1.04 |
| Oncology | 0.96 | 0.60 | 1.54 |
| Pediatric | 0.52 | 0.23 | 1.18 |
| Transplant | 1.22 | 0.76 | 1.97 |
| Vascular | 0.40 | 0.23 | 0.71 |
| Time period | | | |
| March 17–31, 2020 | 0.08 | 0.04 | 0.17 |
| April 1–15, 2020 | 3.13 | 2.33 | 4.22 |
| April 16–30, 2020 | 4.28 | 3.27 | 5.61 |
| May 1–15, 2020 | 2.18 | 1.69 | 2.81 |
| May 16–31, 2020 | 1 [Ref] | | |

DCI = distressed communities index, MIS = minimally invasive surgery.

visit before COVID-19. Video visits at our institution were through a third-party web-based platform. Patients were often, if not exclusively, contacted through MyChart and given a link to an individual surgeon's online portal. Studies examining demographics of MyChart usage have found that lower income patients are significantly less likely to have activated MyChart accounts [29], suggesting that MyChart use is not only reflective of comfort with technology but also socioeconomic status. In addition, patients who paid for their visit with other means —through self-pay, financial assistance, or another mechanism besides government-sponsored or commercial insurance—were over two times as likely to have a telemedicine visit compared

to patients with Medicare. This finding is supported by previous work that found patients with higher copayments and out of pocket costs were more likely to choose a telemedicine visit [22, 24]. Together, the association between greater self-payment and greater telemedicine use may be reflective of greater urgency among patients in seeing a physician, through any means available. Additional research is needed to clarify the reasons behind this paradoxical association.

While associations between socioeconomic status and technological proficiency have been examined in healthcare access, the associations of race/ethnicity with technology remain less clear [22, 24]. Previous work has found that Black patients are significantly less likely to undertake videoconference visits than patients of other races [24]. In contrast, another study found that Black patients were more willing than patients of other races to choose both video and phone visits [22]. The present study, though potentially underpowered, found no difference in telemedicine visits during COVID-19 when stratified by race; this held true when compared to both in-person visits before and during COVID-19. It is likely that social determinants of health, including those not fully captured in this study, drive racial disparities in telemedicine utilization. Visit payor, DCI group, and MyChart status were used as surrogates for patient socioeconomic status and comfort with technology, but these imperfect measures may not adequately reflect differences in access. Although this study did not elucidate differences by race in ability to access telemedicine, it is well known that inequities in access to care are ever-present within the US healthcare system; telemedicine is unlikely to be an exception. Most of the previously mentioned initiatives and funds to improve broadband access by the FCC have been in rural communities, disproportionately benefiting non-Hispanic White Americans [30]. There has been a substantial lack of funding in inner cities, where the majority of communities of color live, despite urban households being three times more likely to not have broadband access than rural households [30]. Qualifications for federal funding exclude nearly all urban centers because they have the infrastructure, but do not take into account accessibility of the people who live there [30]. Given that the population in this study was disproportionately non-Hispanic White and rural, it is quite likely there is a racial divide in telemedicine access nationally that was not captured in this study.

Within this academic surgical department, there was substantial variability in telemedicine utilization by surgical subspecialty. While differing levels of visit urgency may exist between subspecialties, exemplified by the greatest telemedicine uptake among transplant and oncology patients compared to pre-COVID-19 in-person visits, it is likely that surgeon and clinic-dependent factors played a role in the ability to offer telemedicine to patients. Indeed, physician comfort with technology, the ability to make patients aware of telemedicine as an option, and patients being able to see a physician with whom they have established a relationship have been associated with increased telemedicine usage [24, 31, 32]. Though beyond the scope of this study, hospital and physician characteristics appear to have a prominent effect on differences in patient telemedicine adoption.

Telemedicine can be an alternative to the traditional in-patient visit, but this study found that it may not be utilized by the most vulnerable populations. Men and patients of low socioeconomic status, potentially with less access to technology, appear to be disproportionately affected. Given the delays in patient care during the pandemic, these findings are concerning and suggest that some patients may further delay care due to issues or apprehension with telemedicine [8]. Considering many in these populations are already at greater risk of chronic disease, disproportionately affected by the COVID-19 pandemic, and generally have less access to healthcare, it is problematic that telemedicine is possibly another factor contributing to this entanglement of inequality [5, 6, 17]. Strategies for increasing telemedicine use among these at-risk populations are needed. When telemedicine is consistently used and tailored to patient

needs to ensure a successful visit, patients, especially minorities and underserved populations, can benefit [14, 33].

This study has several limitations. Although a wide array of surgical patients were included, the single-center scope restricts generalizability. It is possible that other departments or hospital systems with different patient demographics may exhibit different associations between patient factors and telemedicine utilization. Additionally, we are unable to separate visit types into initial visits versus follow up visits within the current database (non-billable postoperative visits during the global period were not included). Although we would expect similar trends in patient factors across all visit types, it is quite possible that patients attending visits for new appointments and subsequent follow ups are distinctly different, thus confounding the study results. Furthermore, conducting this study during an unprecedented global pandemic may limit the applicability of the findings outside of this context. Further, our data is limited to the initial three months of the pandemic when telemedicine was first implemented and non-urgent in-person visits were rescheduled. These findings may not translate to other timepoints as the pandemic continued. Patient-reported reasoning behind choosing or not choosing telemedicine is an important factor in determining why certain groups of patients have less telemedicine utilization [17, 20, 24, 32]. This study was unable to capture attitudes and perceptions of patients, which if understood, could help further understand disparities in telemedicine access. Finally, patient-reported comfort with technology and digital literacy, instead of proxy measures such as MyChart activation status, may provide a more accurate assessment of this factor [24, 29].

## Conclusion

The COVID-19 pandemic has shown that telemedicine will likely become increasingly necessary to provide safe healthcare. Although it has potential to close the gap in patient access to care, this study found that patient factors influence telemedicine utilization. We cannot rely on telemedicine in its current state to be a solution to equitable access, as it can exacerbate existing disparities for patients from lower resourced communities. Further investigation into underlying barriers to telemedicine access and conscious efforts to mitigate them are required to achieve equity in telemedicine access.

## Supporting information

**S1 Table. Sensitivity analysis excluding pediatric surgery patients of patient characteristics associated with a telemedicine visit during COVID-19, compared to pre-COVID in-person visits.**
(DOCX)

**S2 Table. Sensitivity analysis excluding pediatric surgery patients of patient characteristics associated with an in-person visit during COVID-19, compared to in-person visits before COVID-19.**
(DOCX)

**S3 Table. Sensitivity analysis excluding pediatric surgery patient characteristics associated with a telemedicine visit during COVID-19, compared to an in-person visit COVID-19.**
(DOCX)

## Author Contributions

**Conceptualization:** Courtney M. Lattimore, William J. Kane, Mark A. Fleming, II, Allison N. Martin, J. Hunter Mehaffey, Victor M. Zaydfudim, Shayna L. Showalter, Traci L. Hedrick.

**Data curation:** Mark E. Smolkin, Sarah J. Ratcliffe.

**Formal analysis:** Mark E. Smolkin, Sarah J. Ratcliffe.

**Investigation:** Traci L. Hedrick.

**Methodology:** Courtney M. Lattimore, William J. Kane, Mark A. Fleming, II, Allison N. Martin, J. Hunter Mehaffey, Mark E. Smolkin, Sarah J. Ratcliffe, Victor M. Zaydfudim, Shayna L. Showalter, Traci L. Hedrick.

**Project administration:** Traci L. Hedrick.

**Supervision:** Traci L. Hedrick.

**Visualization:** Courtney M. Lattimore, William J. Kane.

**Writing – original draft:** Courtney M. Lattimore, William J. Kane, Traci L. Hedrick.

**Writing – review & editing:** Courtney M. Lattimore, William J. Kane, Mark A. Fleming, II, Allison N. Martin, J. Hunter Mehaffey, Victor M. Zaydfudim, Shayna L. Showalter, Traci L. Hedrick.

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
