## [Decision Letter · Decision Letter 0]

5 Aug 2021

PONE-D-21-20375

Socioeconomic disparities in telemedicine utilization among surgical patients during COVID-19

PLOS ONE

Dear Dr. Lattimore,

Thank you for submitting your manuscript to PLOS ONE. After careful consideration, we feel that it has merit but does not fully meet PLOS ONE’s publication criteria as it currently stands. Therefore, we invite you to submit a revised version of the manuscript that addresses the points raised during the review process.

We look forward to receiving your revised manuscript.

Kind regards,

Jingjing Qian

Academic Editor

PLOS ONE

Journal Requirements:

Additional Editor Comments:

Please carefully address reviewers' comments. In addition, please describe the sensitivity analysis and its rationale in the Methods section.

Reviewers' comments:

Reviewer's Responses to Questions

**Comments to the Author**

1. Is the manuscript technically sound, and do the data support the conclusions?

Reviewer #1: Partly

Reviewer #2: Yes

2. Has the statistical analysis been performed appropriately and rigorously? 

Reviewer #1: I Don't Know

Reviewer #2: Yes

3. Have the authors made all data underlying the findings in their manuscript fully available?

Reviewer #1: Yes

Reviewer #2: Yes

4. Is the manuscript presented in an intelligible fashion and written in standard English?

Reviewer #1: Yes

Reviewer #2: Yes

5. Review Comments to the Author

Reviewer #1: This is a timely manuscript on an important social issue emerging in telemedicine, an avenue of medicine which continues to be increasing utilized after the pandemic. The authors highlight important social disparities between those who utilized telemedicine during the pandemic versus those who continued with in-person visits. A large retrospective cohort of patients and patient encounters was utilized which is a large strength of the paper. The discussion is well-written and highlights several relevant, up-to-date articles.

There is a conclusion made which I do not believe is sound. Analysis showed that when comparing telemedicine visits during COVID to visits beforehand, those who used telemedicine were more likely to have activated MyChart (lines 163-165). However, when comparing in-person visits during COVID to visits beforehand, those who attended in-person visits were also more likely to have activated MyChart (lines 195-198). Thus regardless of visit type during pandemic (telemedicine vs in-person), the patient was more likely to have activated MyChart. Therefore the conclusion that those more comfortable with technology (using MyChart activation as a gauge) are disproportionately utilizing telemedicine, is false. The statistics need to be re-calculated, or the conclusion withdrawn, or my misunderstanding explained.

My largest critique of this manuscript is the failure to investigate or mention the reason for visit. A patient visit can be of several types - new patient visits, follow-up visits, and post-operative visits to name a few. While routine follow-up visits are easy to perform virtually, some aspects of new patient visits and post-operative visits such as conducting a physical examination, drawing blood, suture removal, dressing change, etc. all necessitate an in-person component. This is potentially massive confounding variable, especially as the manuscript is based entirely on a surgical patient population. For example, perhaps those from more privileged backgrounds were able to cancel their elective surgeries while those from more disadvantaged backgrounds continued with surgery and thus had more in person visits after the pandemic? Or are disadvantaged populations simply more affected from a health perspective by the pandemic (as you cite in your manuscript), requiring more urgent in-person visits? Reason for visit is a vital, possibly confounding variable that ideally be investigated but at minimum mentioned in the discussion.

Minor grammatical errors: line 132 "as well factors" to "as well as factors"; Abstract Methods: "My Chart" to "MyChart" to stay consistent with remainder of manuscript

Reviewer #2: This retrospective cohort study of 14,792 patients seen in an outpatient surgical setting between July 1 2019 to May 31, 2020 compared patient demographic, insurance, SES factors between in-person vs. telemedicine visits prior to and during the COVID-19 pandemic.

Major concerns:

There currently exist a large body of literature describing telehealth use in surgical practices during COVID-19. This paper would add much more insight into telehealth use as restrictions on clinical practice lifted to describe the its use in current contexts.

While institutional policy closed outpatient clinics on March 17th, there were likely changes in healthcare-seeking behavior even before this time as case numbers rose. Would consider another time frame (e.g. same time frame the year before) for a clearer comparison.

Increasingly, it seems that telemedicine in complementary to in-person visits in surgical practices. Type of visit, therefore, is extremely important. What types of visits were these? New patients? Post-op? Return/surveillance patients?

Minor concerns:

How were video visits offered? Using an institutional MyChart based platform? Or whatever communications platform available to the provider/patient (e.g. FaceTime, WhatsApp)? This would give insight into how meaningful MyChart enrollment might be.

Intro:

Line 69 – would change wording to reflect the fact the video visits are not only preferred due to reimbursement – physical examination is possible with video visits (albeit limited) and many non-verbal communication cues can be lost through telephone only assessment

Methods:

- Was “specialty” based on provider or diagnosis? While described in the discussion, would be helpful to be more explicit in the methods.

- Please specify exactly how calendar time was determined. It appears it was considered categorically based on Table 4, but what are the exact definitions?

Results:

- p-values are redundant and not required if presenting 95% CI

- Line 163 – what were these adjusted for? Do all tables represent adjusted odds ratios?

- Table 2 - Would consider using another specialty as a referent as general has one of the smaller n, resulting in increased potential error in OR

- Table 4 – would also give baseline characteristics of “time period” to understand this relationship better.

Discussion:

- Throughout, statements are made based on the results without discussing the comparand population, which can mislead the reader about the conclusions.

6. PLOS authors have the option to publish the peer review history of their article (what does this mean?). If published, this will include your full peer review and any attached files.

Reviewer #1: No

Reviewer #2: No

---

## [Author Response · Author response to Decision Letter 0]

4 Sep 2021

The following responses with a brief letter are also included in the attached file "Response to Reviewers".

Editor Comments:

Please carefully address reviewers' comments. In addition, please describe the sensitivity analysis and its rationale in the Methods section.

• Response: The sensitivity analysis involved rerunning the models while excluding pediatric patients. Considering pediatric patients are younger in age, there were concerns about modeling the age variable and the pediatric surgery category together. This population was inherently dissimilar to the other specialties making it important to determine whether its inclusion was impacting parameter estimates in the model. We found that the model results were similar with or without the pediatric group. This has been added to the Methods section (lines 141-143, highlighted).

Reviewer #1: This is a timely manuscript on an important social issue emerging in telemedicine, an avenue of medicine which continues to be increasing utilized after the pandemic. The authors highlight important social disparities between those who utilized telemedicine during the pandemic versus those who continued with in-person visits. A large retrospective cohort of patients and patient encounters was utilized which is a large strength of the paper. The discussion is well-written and highlights several relevant, up-to-date articles.

There is a conclusion made which I do not believe is sound. Analysis showed that when comparing telemedicine visits during COVID to visits beforehand, those who used telemedicine were more likely to have activated MyChart (lines 163-165). However, when comparing in-person visits during COVID to visits beforehand, those who attended in-person visits were also more likely to have activated MyChart (lines 195-198). Thus regardless of visit type during pandemic (telemedicine vs in-person), the patient was more likely to have activated MyChart. Therefore the conclusion that those more comfortable with technology (using MyChart activation as a gauge) are disproportionately utilizing telemedicine, is false. The statistics need to be re-calculated, or the conclusion withdrawn, or my misunderstanding explained.

• Response: We thank the reviewer for their comment. 54.4% of the patients who attended in-person visits prior to COVID had activated MyChart compared to 61.4% of the telemedicine visits during COVID and 59.3% of in person visits during COVID. The adjusted analysis between in person versus telemedicine visits during COVID was not statistically significant as demonstrated in Table 4. Thus, we removed mention of comfort with technology as a barrier to telemedicine in the conclusion of the abstract and the discussion.

My largest critique of this manuscript is the failure to investigate or mention the reason for visit. A patient visit can be of several types - new patient visits, follow-up visits, and post-operative visits to name a few. While routine follow-up visits are easy to perform virtually, some aspects of new patient visits and post-operative visits such as conducting a physical examination, drawing blood, suture removal, dressing change, etc. all necessitate an in-person component. This is potentially massive confounding variable, especially as the manuscript is based entirely on a surgical patient population. For example, perhaps those from more privileged backgrounds were able to cancel their elective surgeries while those from more disadvantaged backgrounds continued with surgery and thus had more in person visits after the pandemic? Or are disadvantaged populations simply more affected from a health perspective by the pandemic (as you cite in your manuscript), requiring more urgent in-person visits? Reason for visit is a vital, possibly confounding variable that ideally be investigated but at minimum mentioned in the discussion.

• Response: Thank you for bringing up this important question and discussion point. The only visits included in the study were billable visits as the visits were identified from an administrative (financial) database. Postoperative visits within the global (non-billable) period were not included in the database. As far as determining the type of visit amongst the billable visits, we are unable to distinguish between a follow up versus an initial visit in the current database. We agree this is potentially a confounding variable. A statement has been added in the Methods clarifying the visit type (lines 90-93, highlighted) in addition to a further elaboration on this point in the limitations section of the Discussion (lines 330-335, highlighted).

Minor grammatical errors: line 132 "as well factors" to "as well as factors"; Abstract Methods: "My Chart" to "MyChart" to stay consistent with remainder of manuscript

• Response: Thank you for these corrections. The changes have been made and highlighted (line 138 and line 31, respectively).

Reviewer #2: This retrospective cohort study of 14,792 patients seen in an outpatient surgical setting between July 1 2019 to May 31, 2020 compared patient demographic, insurance, SES factors between in-person vs. telemedicine visits prior to and during the COVID-19 pandemic.

Major concerns:

There currently exist a large body of literature describing telehealth use in surgical practices during COVID-19. This paper would add much more insight into telehealth use as restrictions on clinical practice lifted to describe the its use in current contexts.

While institutional policy closed outpatient clinics on March 17th, there were likely changes in healthcare-seeking behavior even before this time as case numbers rose. Would consider another time frame (e.g. same time frame the year before) for a clearer comparison.

• Response: Thank you for this suggestion. Here in central Virginia, our case numbers did not start to rise until late May 2020 as we are in a relatively rural environment. The March 17th shutdown was in response to a decree by the governor largely in part to the national concern and rising cases in Northern Virginia. Although we cannot rule out the possibility that that healthcare-seeking behaviors of our patients may have started to change in the one to two weeks leading up to March 17th date, the number of visits during this short time frame would have been negligible compared to the overall study period. 

Increasingly, it seems that telemedicine in complementary to in-person visits in surgical practices. Type of visit, therefore, is extremely important. What types of visits were these? New patients? Post-op? Return/surveillance patients?

• Response: Thank you for bringing up this important question and discussion point. The study was limited to billable visits. Thus, because postoperative visits occurring during the global period (typically 90 days for most procedures) are not billable, they were not included in the database. As far as determining the type of visit amongst the billable visits, we are unable to distinguish follow up versus initial visits. We agree this is potentially a confounding variable. A statement has been added in the Methods clarifying the visit type (lines 90-93, highlighted) in addition to a further elaboration on this point in the limitations section of the Discussion (lines 330-335, highlighted)

Minor concerns:

How were video visits offered? Using an institutional MyChart based platform? Or whatever communications platform available to the provider/patient (e.g. FaceTime, WhatsApp)? This would give insight into how meaningful MyChart enrollment might be.

• Response: Video visits were offered through an online platform called Doxy.me. Patients were sent a link to their surgeon’s specific online portal through MyChart. MyChart access would have been essential to communicating to patients about accessing the video platform. Although it is possible that patients were sent this link through other means: email, over the phone, etc., we believe MyChart was an integral component to facilitating video appointments. We have added a statement addressing this point in the Methods (lines 108-110, highlighted) and Discussion (lines 269-271, highlighted).

Intro:

Line 69 – would change wording to reflect the fact the video visits are not only preferred due to reimbursement – physical examination is possible with video visits (albeit limited) and many non-verbal communication cues can be lost through telephone only assessment

• Response: Thank you for this important addition. A change has been made in the Introduction (lines 67-70, highlighted).

Methods:

- Was “specialty” based on provider or diagnosis? While described in the discussion, would be helpful to be more explicit in the methods.

• Response: We have aimed to make this clearer in the Methods by adding further explanation. (line 106, highlighted).

- Please specify exactly how calendar time was determined. It appears it was considered categorically based on Table 4, but what are the exact definitions?

• Response: Thank you for bringing this to our attention. We have added the exact dates for the time periods into Table 4 (highlighted) and have also stated these dates within the study variables section of the Methods (lines 120-121, highlighted).

Results:

- p-values are redundant and not required if presenting 95% CI

• Response: We have removed p-values from Tables 2-4 and our supplemental tables S1-S3.

- Line 163 – what were these adjusted for? Do all tables represent adjusted odds ratios?

• Response: Each table represents one model and all of the model variables. The odds ratios within the table are adjusted for all of the variables in the model. We have clarified this in line 172 (highlighted).

- Table 2 - Would consider using another specialty as a referent as general has one of the smaller n, resulting in increased potential error in OR

Response: The ‘general’ category was chosen as the reference category because it was deemed the most meaningful to display comparisons with the other specialties. In our opinion, had we chosen a different reference category with a larger n, the comparisons displayed in the table would likely have been of lesser interest.

- Table 4 – would also give baseline characteristics of “time period” to understand this relationship better.

• Response: We have added the exact dates for the time periods into Table 4 and have also stated these dates within the study variables section of the Methods (lines 120-123, highlighted).

Discussion:

- Throughout, statements are made based on the results without discussing the comparand population, which can mislead the reader about the conclusions.

• Response: Thank you for this feedback. Several changes have been made and highlighted throughout the Discussion to address the comparand group of results on which the discussion is based.

---

## [Decision Letter · Decision Letter 1]

16 Sep 2021

PONE-D-21-20375R1Socioeconomic disparities in telemedicine utilization among surgical patients during COVID-19PLOS ONE

Dear Dr. Lattimore,

Thank you for submitting your manuscript to PLOS ONE. After careful consideration, we feel that it has merit but does not fully meet PLOS ONE’s publication criteria as it currently stands. Therefore, we invite you to submit a revised version of the manuscript that addresses the points raised during the review process.

We look forward to receiving your revised manuscript.

Kind regards,

Jingjing Qian

Academic Editor

PLOS ONE

Journal Requirements:

Additional Editor Comments (if provided):

The authors properly addressed majority of the comments provided. However, a few additional edits need to be made to ensure accuracy of results reporting and conclusion drawing.

1. Abstract: page 2, Conclusions. Based on your findings, the current conclusions are not appropriately supported, because the significance of DCI (table 2) and female (table 4) came from two different models with different comparison groups. Please consider to use the revised Conclusions "During the first 3 months of the COVID-19 pandemic, Telemedicine was more likely used by females patients and those without government or commercial insurance compared to patients who used in-person visits. Interventions using telemedicine to improve health care access might consider such differences in utilization." Given this change, I suggest the authors to remove "socioeconomic" in their title.

2. Discussion, page 14, line 227-229: please consider revising this sentence to "Overall, telemedicine was more likely utilized by those without government or commercial insurance and women."

3. Limitations, page 19. The authors only had access to less than 3 months of data after the COVID-19 outbreak, from March 17, 2020 to May 31, 2020. Please add a limitation of the early observation of the association based on the short period availability of data, which might be different from later into the pandemic.

4. Conclusion, page 19, line 347. Please consider to change the sentence to "this study found that patient factors influence telemedicine utilization."

Reviewers' comments:

Reviewer's Responses to Questions

**Comments to the Author**

1. If the authors have adequately addressed your comments raised in a previous round of review and you feel that this manuscript is now acceptable for publication, you may indicate that here to bypass the “Comments to the Author” section, enter your conflict of interest statement in the “Confidential to Editor” section, and submit your "Accept" recommendation.

Reviewer #1: All comments have been addressed

2. Is the manuscript technically sound, and do the data support the conclusions?

Reviewer #1: Yes

3. Has the statistical analysis been performed appropriately and rigorously? 

Reviewer #1: Yes

4. Have the authors made all data underlying the findings in their manuscript fully available?

Reviewer #1: Yes

5. Is the manuscript presented in an intelligible fashion and written in standard English?

Reviewer #1: Yes

6. Review Comments to the Author

Reviewer #1: (No Response)

7. PLOS authors have the option to publish the peer review history of their article (what does this mean?). If published, this will include your full peer review and any attached files.

Reviewer #1: No

---

## [Author Response · Author response to Decision Letter 1]

23 Sep 2021

Journal Requirements:

• The reference list was reviewed in its entirety and is both complete and correct with no retracted articles cited. A corrected citation was made in the Introduction (line 64, highlighted). 

Additional Editor Comments (if provided):

The authors properly addressed majority of the comments provided. However, a few additional edits need to be made to ensure accuracy of results reporting and conclusion drawing.

1. Abstract: page 2, Conclusions. Based on your findings, the current conclusions are not appropriately supported, because the significance of DCI (table 2) and female (table 4) came from two different models with different comparison groups. Please consider to use the revised Conclusions "During the first 3 months of the COVID-19 pandemic, Telemedicine was more likely used by females patients and those without government or commercial insurance compared to patients who used in-person visits. Interventions using telemedicine to improve health care access might consider such differences in utilization." Given this change, I suggest the authors to remove "socioeconomic" in their title.

• Thank you for bringing this important point to our attention. The changes to the abstract have been made (lines 42-45, highlighted) and “Socioeconomic” has been removed from the title.

2. Discussion, page 14, line 227-229: please consider revising this sentence to "Overall, telemedicine was more likely utilized by those without government or commercial insurance and women."

• Thank you for this suggestion. The revision to this sentence has been made (lines 226-227, highlighted)

3. Limitations, page 19. The authors only had access to less than 3 months of data after the COVID-19 outbreak, from March 17, 2020 to May 31, 2020. Please add a limitation of the early observation of the association based on the short period availability of data, which might be different from later into the pandemic.

• We agree that the three months of data at the beginning of the pandemic are indeed a limitation. A statement to reflect this has been added (lines 334-337, highlighted).

4. Conclusion, page 19, line 347. Please consider to change the sentence to "this study found that patient factors influence telemedicine utilization."

• This change has been made. (line 347, highlighted)

---

## [Editor Report · Decision Letter 2]

28 Sep 2021

Disparities in telemedicine utilization among surgical patients during COVID-19

PONE-D-21-20375R2

Dear Dr. Lattimore,

We’re pleased to inform you that your manuscript has been judged scientifically suitable for publication and will be formally accepted for publication once it meets all outstanding technical requirements.

Kind regards,

Jingjing Qian

Academic Editor

PLOS ONE

Additional Editor Comments (optional):

Thanks for making the suggested editorial edits.
---

## [Editor Report · Acceptance letter]

30 Sep 2021

PONE-D-21-20375R2 

Disparities in telemedicine utilization among surgical patients during COVID-19 

Dear Dr. Lattimore:

I'm pleased to inform you that your manuscript has been deemed suitable for publication in PLOS ONE. Congratulations! Your manuscript is now with our production department. 

Kind regards, 

on behalf of

Dr. Jingjing Qian 

Academic Editor

PLOS ONE